# D-Limonene as a Promising Green Solvent for the Detachment of End-of-Life Photovoltaic Solar Panels under Sonication

Dina Magdy Abdo [1], Teresa Mangialardi [2], Franco Medici [2] and Luigi Piga [2,*]

1   Central Metallurgical Research and Development Institute, P.O. Box 87, Helwan, Cairo 11421, Egypt;
    eng.dina_magdy@yahoo.com
2   Department of Chemical Engineering, Materials and Environment, Sapienza University of Rome,
    Via Eudossiana 18, 00184 Roma, Italy; teresa.mangialardi@uniroma1.it (T.M.);
    franco.medici@uniroma1.it (F.M.)
*   Correspondence: luigi.piga@uniroma1.it

**Abstract:** Consumption of photovoltaic solar panels is expected to increase, so the growing amount of end-of-life (EOL) solar panels will require large spaces for their disposal, which at the moment costs around 200 euros/ton. Thus, a proper treatment technique to recover secondary materials from this waste, which are mainly copper, aluminum, silicon, high-transmittance glass, and plastics, must be developed. The last three components are strongly attached to each other; hence, their detachment is necessary for recovery. To achieve this objective, a chemical route was chosen; in fact, solvent extraction is highly recommended, as it has a high separation efficiency. In this study, D-limonene as a bio-solvent was examined for detaching different components of solar panels from each other. A high efficiency for ethylene vinyl acetate (EVA) dissolution and components' detachment under different conditions was achieved with the help of sonication power. The effects of sonication power, thermal pre-treatment, temperature, and contact time on detachment percentage were examined, and the best conditions (namely, no pre-treatment, medium sonication power of 450 W, temperature of 60 °C, and a contact time of 120 min) were found for total component detachment. Additionally, the recyclability of D-limonene was examined, and it was established that the solvent could carry out 100% component detachment for three cycles.

**Keywords:** photovoltaic solar panels; recycling; treatment; D-limonene; ethylene vinyl acetate (EVA); detachment

## 1. Introduction

The energy demand has been growing during the last few years due to the rapid growth in the global population; thus, the consumption of energy has increased. Fossil fuels are the most traditional energy source used but are non-sustainable and cause several environmental problems and health issues due to the emission of greenhouse gases [1,2]. Based on this evidence, using renewable solar power as a green sustainable energy source has attracted most researchers' attention. Moreover, the World Energy Council favors the replacement of fossil fuels with solar power as an alternative green energy source [3]. Therefore, the consumption of photovoltaic solar panels (PV) is expected to significantly increase in the near future. However, considering that PV panels have a lifetime of about 25 years, their widespread use will become a more and more serious problem day by day. At the end-of-life (EOL), photovoltaic solar panels belong to the waste from electrical and electronic equipment (WEEE), of which global production is expected to reach 60–78 million tons by 2050 [4,5]. Thus, proper treatment methods to manage and recycle this equipment are needed to avoid the health and environment problems associated with the disposal of such waste [6,7]. Research in this regard is active across the world [8–11]. Moreover, the EU-WEEE Directive (2018/19/EU) requires that after 2018, 80% of the weight of PV panels

be recycled and 85% recovered. Hence, increasing research and development is required to meet the EU requirements.

Silicon solar panels are the most common type used globally, where about 90% of panels present in the market have the same composition [12]. As shown in Figure 1, the panel consists of five layers that are chemically bonded with ethylene vinyl acetate copolymer (EVA). To recycle EOL-PV panels, a suitable method must be achieved to detach the layers efficiently.

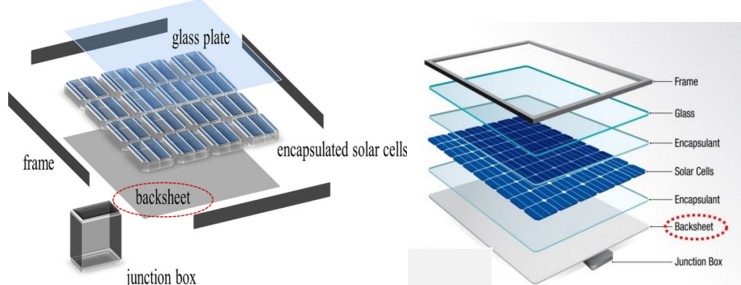

**Figure 1.** Saur Energy International, solar panel structure.

The main basic functions of an encapsulating material such as EVA are the following:

- To provide a structural support and positioning in the module-design layout;
- To achieve and maintain optical coupling between the solar cell and the glass, keeping the incidence of solar radiation at least 90% during the years of operation;
- To maintain the physical isolation of solar cells and components and to protect the circuit under operative conditions;
- To achieve and protect the electrical insulation between the solar cells and circuit elements during the operating life of the photovoltaic modules;
- To maintain the integrity of the electric circuit, which generates the required current and voltage under light exposure; moreover, the transparency of the encapsulant must also be maintained throughout the operative lifetime of the photovoltaic modules.

Different common methods are used to separate these layers [13–20], as shown in Table 1, including thermal treatment, physical treatment, and chemical treatment either using chemical processing with organic acids or using a solvent extraction technique. Among the chemical methods, the solvent extraction technique is highly economically recommended, as it could achieve high separation efficiency with low operating temperatures [8].

**Table 1.** Several treatment types for EOL solar panels.

| Method | Advantage | Disadvantage | Reference |
|---|---|---|---|
| Thermal treatment | • Separation efficiency<br>• Fast separation | • Harmful gases evolved<br>• Specific equipment required<br>• High energy consumption | [13,14] |
| Physical treatment | • Environmentally method<br>• Fast separation<br>• Simple operation | • Non-efficient separation<br>• Low recovery rates<br>• Damage to solar cells | [15–17] |
| Chemical treatment | • Simple operation<br>• Low energy consumption<br>• Fast separation<br>• Separation efficiency<br>• Low cell damage | • Long reaction time<br>• Toxic chemical byproducts | [18–20] |

The most important step in the separation process is to reduce the adhesion between the panel layers, and this can be achieved by completely dissolving the encapsulating layer of EVA. This component first absorbs the suitable solvent, then it swells, causing weakness to the bond, and then it finally detaches from back sheet. In choosing the solvent, it must be taken into account that it must not react with the back sheet.

Different types of plastic are used as a back sheet, but the most commonly used is polyvinyl fluoride (PVF) [21]. Most solvents that have been used in PV panel layers detachment are organic solvents, especially toluene, whose use was recently restricted by REACH (Registration, Authorisation of Chemicals, EC Regulation n. 1907/2006), as it is considered a hazardous material due to its flammability and toxicity [22]. The green solvent extraction method is to use bio-solvents derived from natural renewable sources or wastes, which are biodegradable and less toxic than most organic solvents. D-limonene solvent can replace many undesirable solvents such as benzene, hexane, trichloroethylene, toluene, and xylene [15]. It is extracted mainly from natural fruits, and it is a nonpolar, water-insoluble solvent and shows great results in the application of the solvent extraction technique as a substitute for petroleum derivatives for the recovery of natural products [23] or discarded plastics [24]. As the D-limonene molecule contains a benzene ring (non-polar), and the polarity of the vinyl acetate group is also weak, the affinity between two chemical groups is large, so it easily dissolves EVA, and thus, the solar panel back sheet may be efficiently detached.

The main aim of the present study is to examine the effect of D-limonene in the detachment of EVA and separation of the back sheet layer of PV panels while also finding the suitable conditions for the total detachment of the back sheet. For this purpose, different operative parameters, including pre-treatment effect, extraction time, sonication power, and extraction temperature, were considered. In addition, based on economic and environmental considerations, the recyclability of used limonene was investigated for the selected parameters.

A recent study [25] showed that at present, recycling is not convenient from an economic point of view, but it is from an environment point of view. Analysis of the economic scenario showed that improvements in technology, expansion of the recycling scale, reducing transport distances, and subsidization are all effective measures for enhancing the total benefits of PV-waste recycling.

## 2. Materials and Methods

### 2.1. Sample Preparation

The end-of-life (EOL) solar panels used during this work were supplied by the municipality of Celano (L'Aquila, Italy), where Europe's largest photovoltaic park, which is owned by a public administration, is installed. The park consists of 61,000 PV modules (polycrystalline–silicon type) with a power of 330 W each, and it spreads across a field of 28 ha. The end-of-life polycrystalline–silicon PV panels were treated by first applying a physical-mechanical treatment followed by a chemical one. The modules were preliminarily manually dismantled to recover the external aluminum frame; moreover, cables and junction boxes were also detached. Then, test samples consisting of tightly adhered plastic back sheet, glass, solar cells, and encapsulant were prepared.

The test sample preparation was achieved in two stages: firstly, one panel was divided into $13 \times 13$ cm$^2$ homogenous plates by means of a circular saw and then cut manually into pieces of about $5 \times 3$ cm$^2$, as shown in Figure 2. Half of the samples were heat-pretreated for 12 h in a laboratory oven at 200 °C (Vittadini, Italy) to examine the effect of thermal pre-treatment on the detachment. Each $5 \times 3$ cm$^2$ piece was weighted before the detachment tests.

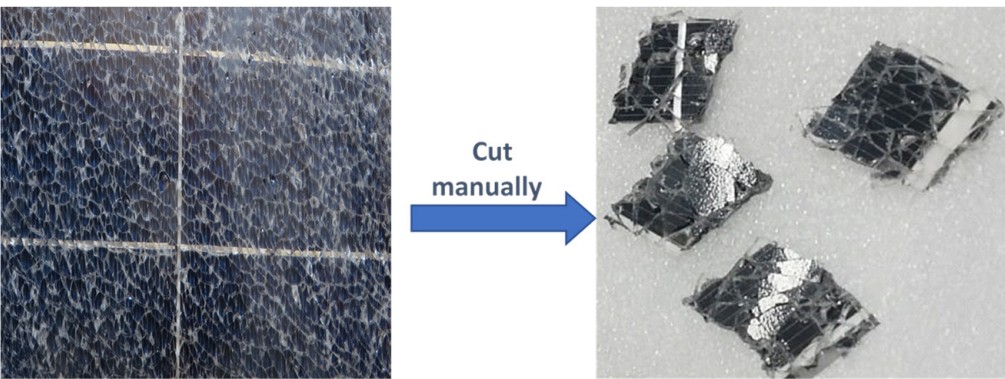

**Figure 2.** Sample preparation.

### 2.2. Detachment Procedures

D-limonene (Sigma-Aldrich, chemical grade 98%) with a normal boiling point of 176 °C and density of 0.85 g/mL was used as a bio-solvent without any further addition or treatment.

The detachment process was achieved by sinking the sample horizontally in 100 mL solvent within a 250 mL conical glass flask.

Preliminary tests were carried out utilizing samples of five different dimensions, namely $4 \times 3$, $5 \times 3$, $6 \times 3$, $5 \times 4$, and $8 \times 4$ cm$^2$. These preliminary tests showed that the percentage of detachment did not depend on the size of the sample within the limit of $+/-5\%$, which is the uncertainty of measurement.

Then, a set of three pieces of about $5 \times 3$ cm$^2$, equivalent to an average weight of 10 g, was used for each test so that the average sample-to-solvent weight ratio was equal to 1:8.5. To avoid using deficient or excess solvent, it was preferred to immerse the sample horizontally rather than vertically, as pressure is homogenous across the horizontal position [26]. The conical flask containing the sample and solvent was then placed in an ultrasonic equipment with a variable external power supply in order to examine the effect of different power on the detachment of solar panels layers.

A GT Sonic T27 (Shenzhen, China) ultrasound machine equipped with a digital controlled timer, adjustable heating (0–120 °C), and variable power function (0–900 W) was utilized during the experimental tests.

Different parameters were examined during EVA dissolution and the detachment of back sheet from EOL solar panels, including the temperature, which varied in a range significantly lower than the solvent boiling point (25 °C, 60 °C, and 80 °C) to avoid its thermal degradation; the contact time (10 min, 50 min, and 120 min); and the sonication power (200 W, 450 W, and 700 W). All the mentioned parameters were examined on both thermally pre-treated and not-pre-treated samples. At the established contact times, the pieces of PV panel were extracted from the solvent, dried at 80 °C, and then weighted in order to calculate the percentage of detachment.

Each detachment test was repeated three times, and the overall results are reported in Figure 3 as the mean value of three determinations. The deviation of the minimum and maximum obtained values with respect to the average was always less than 5%.

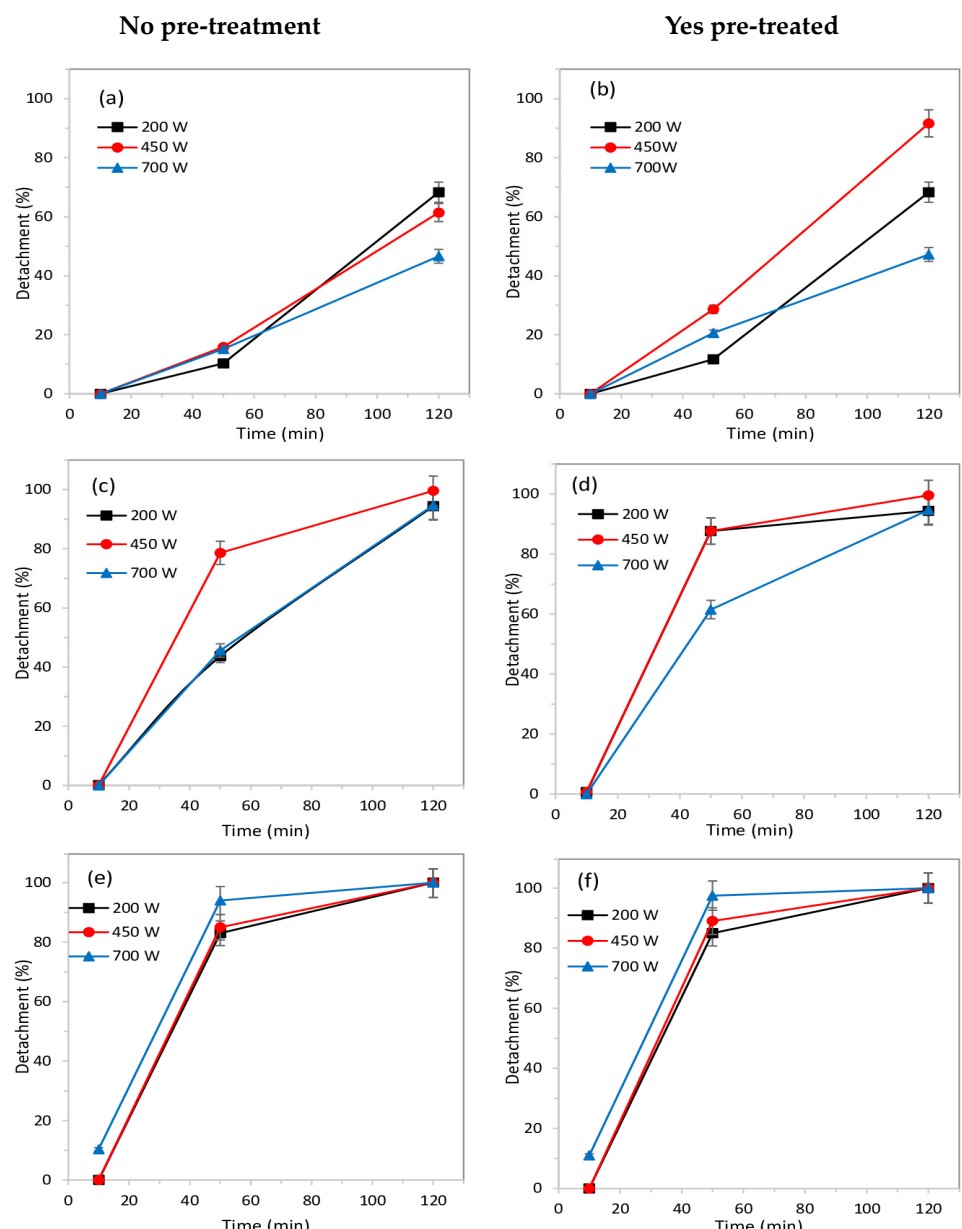

**Figure 3.** Effect of temperature and thermal pre-treatment on back sheet detachment as a function of time and sonication power: 25 °C (**a**,**b**); 60 °C (**c**,**d**); 80° C (**e**,**f**).

### 2.3. Calculation

The different values of the sonication power applied to test samples were achieved by operating the ultrasonic equipment at constant voltage of 250 V and variable intensities of current (i.e., 0.8 A, 1.8 A, and 2.8 A). Equation (1) was applied to calculate the sonication power:

$$Power = Voltage (V) \times Current (A) \tag{1}$$

The percentage of detachment was calculated according to the following equation:

$$\% \quad Detachment = \frac{Wt_i - Wt_F}{Wt_i - Wt_b} \times 100 \tag{2}$$

where
$Wt_i$: initial weight of the sample;
$Wt_F$: final weight of the sample after detachment;
$Wt_b$: weight of back sheet (2.5% of initial sample weight).

## 3. Results and Discussion

Thermal pre-treatment mainly acts on the EVA layer, and it was observed that thermal pre-treatment has a significant positive effect on the EVA dissolution and back sheet detachment when applying a low-temperature (25 °C) solvent treatment and a medium sonication power (450 W). As observed from Figure 3a,b, the detachment increase for pre-treated samples with respect to non-pre-treated ones is about 80% and 50% after 50 and 120 min, respectively. On the other hand, as temperature increased and reached 60 °C, the effect of thermal pre-treatment was significant for a contact time of 50 min at the low sonication power (about 100% increase in the detachment with thermal pre-treatment) and moderate for medium and high powers (11–34% increase in detachment). The effect became negligible as testing time increased to 120 min. As shown in Figure 3c,d, the same behavior was observed for both treated and untreated samples regardless of the sonication power value.

By increasing the temperature to 80 °C, the predominant role of the temperature becomes clear. Over the time range investigated, from 10 to 120 min, the results obtained from pre-treated or not-pre-treated samples are comparable, as represented in Figure 3e,f.

Increasing the sonication power directly affects the molecular collision speed due to increasing the rate of cavitation; thus, the applied energy increases, causing breaking for the chains of the EVA polymer, all of which leads to complete detachment of the back sheet [27]. These criteria explain the phenomena shown in Figure 3a–d, as it was observed that the medium power of 450 W (red line representation) facilitated better detachment than the low power of 200 W (black line representation) at temperature ranging from 25 °C to 60 °C for both pre-treated and non-pre-treated samples.

Pre-treatment positively affects the detachment process at low power and low temperature, while 100% detachment can be achieved for both pre-treated and non-pre-treated samples by applying 450 W. Moreover, 100% detachment was achieved by applying 450 W for 120 min.

On the other hand, applying the high power of 700 W, as shown in Figure 3a–d (blue line representation), negatively affects the detachment process in the temperature range of 25 °C to 60 °C. This may be due to the fact that as sonication power increases, more cavitation bubbles are produced, and their tendency to collide increases as well. This phenomenon will require more time to break down bubbles, thus reducing the ultrasonic positive effect and causing insufficient detachment [28]. As revealed in Figure 3e,f, increasing the temperature to 80 °C overcomes the tendency of bubble cushion formation; thus, by increasing the power, the cavitation increases, and only in this case is the percentage of detachment directly proportional to the sonication power applied before reaching 120 min.

As observed overall from Figure 3, increasing temperature has a significant positive effect on the detachment process. However, at 60 °C, when applying the medium sonication power of 450 W for 120 min both on pre-treated and not-pre-treated samples (Figure 3c,d), the back sheet detaches completely. Furthermore, at 80 °C for 120 min, detachment is total for both pre-treated and non-pre-treated samples (Figure 3e,f), and this can be attributed to the enhancement in the separation effect due to the increase in molecular thermal motion as the temperature increases.

It may also be concluded from the achieved results that sonication contact time is the fundamental parameter affecting the detachment percentage, and 120 min is required to achieve total detachment.

*Detachment Condition*

Using organic solvents, the EOL components were separated by dissolving the encapsulant EVA layers in the solvent, causing the back sheet's detachment. The main reason for the dissolution of EVA is the diffusion of the solvent (D-limonene) in the polymer, which leads to the disruption of polymer bonds and misalignment of broken chains.

Table 2 summarizes all the conditions applied in this study to achieve 100% detachment using D-limonene as the solvent. An image of a completely detached sample is

shown in Figure 4. Thus, by achieving 100% detachment, the recovery of various valuable components can be easily facilitated.

**Table 2.** Summarization of conditions that allowed a total back sheet detachment.

| Sonication Power | 100% Detachment Condition | Pre-Treatment | Main Affecting Parameter |
|---|---|---|---|
| Low (200 W) | Temperature 80 °C for 120 min | No effect | Temperature |
| Medium (450 W) | Temperature 60 °C for 120 min | No effect | Time |
| High (700 W) | Temperature 80 °C for 120 min | No effect | Time |

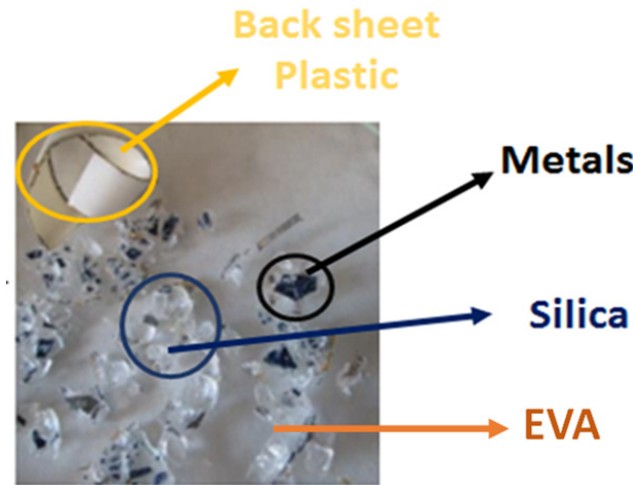

**Figure 4.** Photo of treated solar panels with a 100% detached back sheet. Condition: T = 60 °C, no pre-treatment, and 120 min under 450 W ultrasonic power.

Solvent extraction is an efficient process for back sheet detachment from EOL solar panels, and in this regard, several studies have been performed using different solvents (hexane, trichloroethylene, o-dichlorobenzene, butanone, and toluene) and under different conditions; some of these studies are summarized in Table 3. The main problem arising from the use of organic solvents is their toxicity; in fact, all the organic solvents listed in Table 3 are hazardous for the environment and for health. The commercial prices in Europe vary from about 0.97 USD/kg (toluene) to about 2.1 USD/kg (trichloroethylene), but environmental reasons (bio-degradability) make limonene preferable to other solvents tested, even if its price is about 5 USD/kg. The dissolution of the EVA layer basically depends on its bond and stickiness: if the layer is very adherent, more severe conditions are required to spread the solvent through the adhesive system. In this regard, Figure 5 shows the different percentages of back sheet detachment under different residence times.

**Table 3.** Comparison between the back sheet detachment achieved in the current work and in other works.

| Solvent Used | Applied Conditions | Detachment | Ref. |
|---|---|---|---|
| D-limonene | T = 60 °C, no pre-treatment, 120 min, and power 450 W | 100% | Current study |
| Hexane | T = 70 °C, ultrasonication for 15 min, for 24 h | 92.4% | [29] |
| Trichloroethylene | T = 70 °C, with a module to solvent ratio of 1:7.44, and in a horizontal position for 8 h | 100% | [26] |
| o-Dichlorobenzene | T = 120 °C for one week and with pre-treatment at 155 °C for 15 min | 100% | [30] |
| Butanone | T = 70 °C for 210 min and high sonication power with 50% concentration | 100% | [31] |
| Toluene | T = room temperature for 3 days | 100% | [32] |

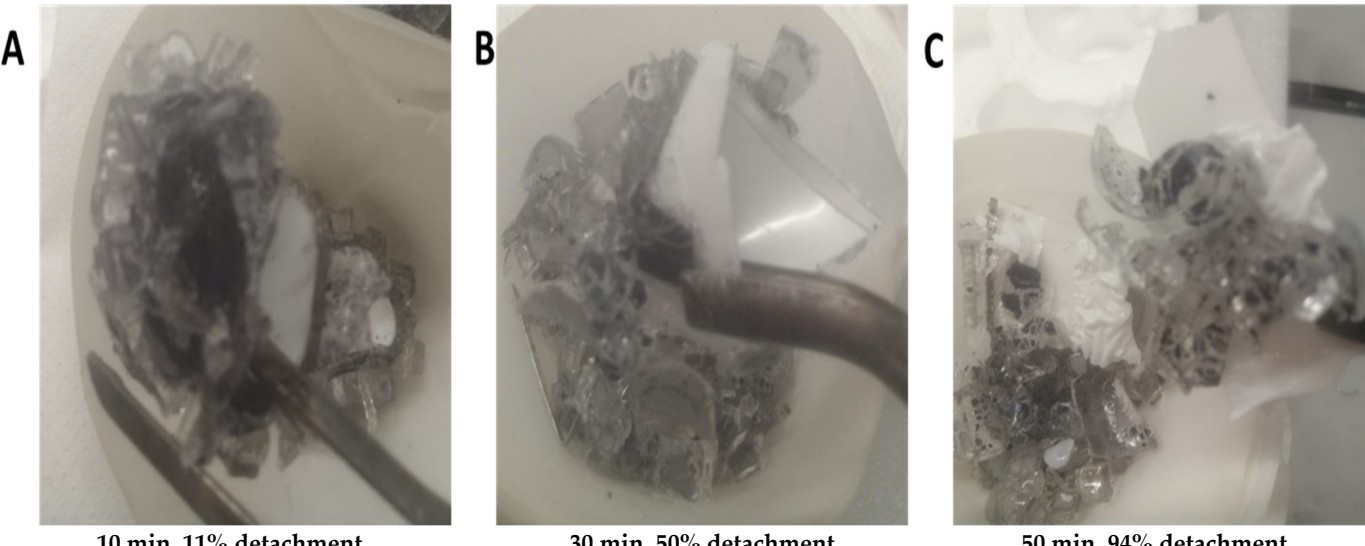

**10 min, 11% detachment**   **30 min, 50% detachment**   **50 min, 94% detachment**

**Figure 5.** Different back sheet detachments under different residence times. No pre-treatment, T = 80 °C, and ultrasonic power 700 W.

## 4. Solvent Recyclability

To verify the possibility of using the proposed process, once the best experimental conditions were identified, i.e., no pre-treatment, medium sonication power (450 W), and temperature of 60 °C for 120 min, these conditions were chosen to examine the recyclability of the solvent for more than one cycle.

Therefore, five extraction tests were performed in a series, using samples of PV solar panels obtained as described in paragraph 2.1. The first of these samples was treated with fresh D-limonene, while the subsequent ones were subjected to extraction using as a solvent the solutions resulting from the treatments of the samples that preceded them in the series.

It was found that during the detachment process, the encapsulant (EVA) dissolves in D-limonene, allowing 100% detachment of the back sheet, for three cycles, after which the disjunction rate gradually decreases with each serial cycle, as shown in Figure 6.

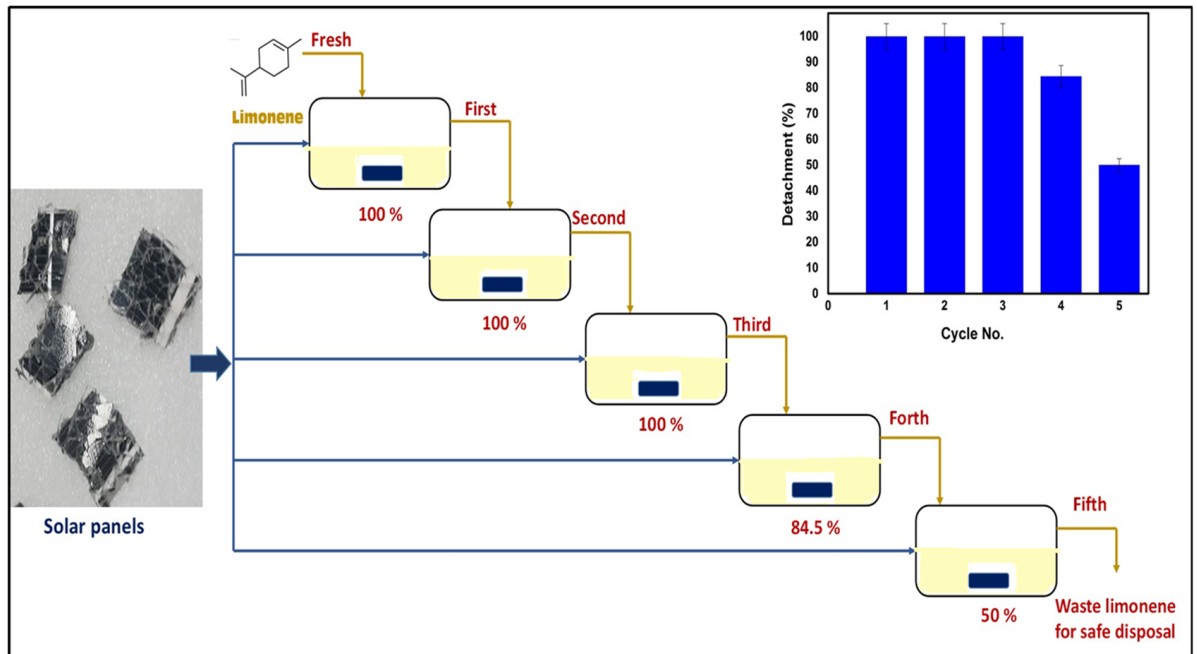

**Figure 6.** Recyclability of D-limonene as a green solvent for solar panels' detachment.

This is because the EVA concentration gradually increases until the solvent approaches the saturation stage, at which the EVA can no longer dissolve. This means that the solution of EVA in D-limonene will have to be purified beyond three cycles so that the solvent can be reused.

EVA dissolution may be considered as controlled by four main factors: (a) the type of solvent, (b) the operating temperature that contributes according to the thermodynamics and the kinetics of the process, (c) the surface area exposed to the solvent, and (d) the agitation/mixing conditions (mass-transfer parameter). The advantage of the proposed process is to use a bio-degradable solvent, in contrast with other processes, and described in the literature [26,27,30–32], in which organic solvents that are impacting the environment are used.

However, the process needs to be optimized both for the effect of the surface area exposed to the solvent and to fully understand the number of possible sequential treatment cycles. In the last regard, the type of process and the appropriate operational conditions will also have to be identified to quantitatively recover the solvent from the recycled extraction solutions.

Finally, to explain the entire treatment process utilized in the laboratory, Figure 7 is shown. Figure 7 highlights the different stages of treatment and the effect of the final detachment and the separation of the different components (metals, silica, and back sheet) [33].

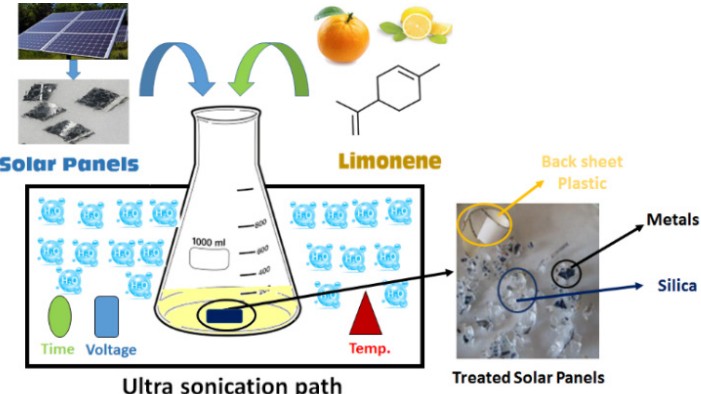

**Figure 7.** Process scheme used in the laboratory.

In the current state of knowledge and technical development, it is difficult to evaluate the economic return of the overall recovery operations of the components present in the photovoltaic panels. In fact, the price of the production and installation of solar panels has been decreasing in recent years. Moreover, the size of the PV module recycling plant has a significant influence on its profitability [34].

Some published works [34,35] show the profitability limit for 10,000 tons/year of treated waste, while other works [36,37] fix the minimum value at 20,000 tons/year. Considering the different disposal scenarios, it is clear that the revenues obtained from the whole recovery of the components do not compensate for the investment and operating costs using small-scale plants in the absence of an environmental contribution linked to the recovery of this waste.

Like other electronic wastes, PV modules can have an adverse impact on the environment if delivered directly to a landfill, as the toxic heavy metals (cadmium, copper, selenium, and tin) that they contain can leach into the groundwater through the soil. In order to avoid recurrence of negative experiences surrounding the final disposal of electrical and electronic wastes, it becomes essential to organize management of the EOL solar panels through combined mechanical and chemical recovery systems. The entire recovery process must take into account that the overall mass composition of the recovered materials for a polycrystalline PV panel is approximately distributed as follows: plastic 9.93 (wt %), valuable metals 1.37 (wt %), glass 84.02 (wt %), and other metals 0.71 (wt %). The presented

mass balance does not equal 100%: this is due to the inevitable losses during each phase, in particular crushing and milling/sieving [15,34].

## 5. Conclusions

D-Limonene is a bio-solvent and was selected as a layer-separation reagent based on its molecular structure, which is similar to the structure of EVA. Its good wettability on the glass makes it easier penetrate the glass–EVA film gap, and its high boiling point (176 °C) provides a wide range of applications.

This work demonstrates that D-limonene is effective for separating the back sheet of end-of-life photovoltaic solar panels under ultrasonic irradiation for three use cycles.

The most suitable conditions for separation were studied by examining the effect of pre-treatment, varying temperature, residence time, and ultrasonic power.

Based on the experimental tests, the following conclusions were reached:

(1) The effect of temperature is strongly effective above 25 °C, and in fact, at this value of temperature (25 °C) and for a contact time lower than 50 min, the reaction yields are lower than 40%;

(2) At low sonication power (200 W), the thermal pre-treatment has no effect on the EVA dissolution, and the temperature is the main parameter influencing the process;

(3) At an average sonication power of 450 W, the pre-treatment has no effect on EVA dissolution, and time is the main parameter influencing the process;

(4) A temperature of 60 °C and 120 min of contact time are required to obtain total detachment regardless of the sonification power;

(5) With a high sonification power of 700 W, the pre-treatment does not have a positive effect, and the residence time is the main parameter influencing the process. Furthermore, separations of more than 80% can be obtained in 50 min of contact time both at 60 °C and at 80 °C.

A steady increase in end-of-life polycrystalline silicon photovoltaic panels requires the development of recycling technologies to guarantee sustainable environmental management and a circular economy. Finally, for a large-scale application of the proposed process for the recovery of photovoltaic modules, further intensive studies must be carried out regarding the use of bio-degradable solvents such as D-limonene.

**Author Contributions:** Conceptualization, D.M.A., F.M. and L.P.; methodology, D.M.A., F.M. and L.P.; validation, D.M.A., T.M., F.M. and L.P.; formal analysis, D.M.A., F.M. and L.P.; investigation, D.M.A., F.M. and L.P.; resources, F.M. and L.P.; data curation, D.M.A. and F.M; writing—original draft preparation, D.M.A.; writing—review and editing, D.M.A., T.M., F.M. and L.P.; supervision, T.M., F.M. and L.P.; project administration, D.M.A., F.M. and L.P.; funding acquisition, F.M. and L.P. All authors have read and agreed to the published version of the manuscript.

**Funding:** This research was funded by "Sapienza" University of Roma, grants "R.S. 2018, PIG_18" and "R.S. 2019, MED_15".

**Data Availability Statement:** Not applicable.

**Acknowledgments:** The authors thank the Municipality of Celano (L'Aquila, Italy) for partial economic support for research and for the availability of photovoltaic panels and Mauro Ferrini and Antonio Scoppettuolo for their suggestions during the experimental tests.

**Conflicts of Interest:** The authors declare no conflict of interest.

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
