# Peer review of "D-Limonene as a Promising Green Solvent for the Detachment of End-of-Life Photovoltaic Solar Panels under Sonication"

_processes, doi:10.3390/pr11061848_

Round 1

Reviewer 1 Report

Many crucial details are missing in the work, and it will take a significant amount of work before it can be considered for publication. 

This includes changing the sample size and scaling it up for at least one panel mass.

The cost of the power used to conduct each experiment must be carefully calculated.

The carbon footprint of each set of experiments should be calculated.

Reviewer 2 Report

The options you took in your experiments are narrow-ranged, but your results and conclusions could be useful in the detachment of the PV solar panels.

your work is good, but you need to check the followings:

1. In your introduction mentioned that the waste from electric and electronic equipment is expected to reach 60-78 million tons in 2025, so if we will use your proposed method, what are the advantages (Cost, Capacity, impact on the Environmental)?

2. I suppose to use a wider range of temperatures in your experiment.

3. In sample preparation, it is better to mention some explanation and justification about the size of the sample used.

4. You need additional figures showing the process of your detachment procedures.

5. Line No. 182. Change the symbol %.

Reviewer 3 Report

In this work, D-Limonene was studied as a bio-solvent for separating different components of solar panels from each other. High efficiency was achieved in dissolving ethylene vinyl acetate (EVA) and detaching the components under various conditions with the assistance of sonication power. The impact of sonication power, thermal pretreatment, temperature, and contact time on the detachment percentage was investigated, and the optimal conditions were determined for complete component detachment. Additionally, the recyclability of D-Limonene was examined, and it was found that the solvent could accomplish 100% component detachment for three cycles.

This study provides valuable insights that make it suitable for publication in processes (MDPI) journal. However, I would like to offer the following comments and suggestions to further enhance the manuscript:

1.     The introduction would benefit from more references to enhance the reader's understanding of the subject and place the current study in the context of existing research. Additional references will establish a strong foundation for the significance and novelty of this research, ensuring its alignment and contribution to the existing body of knowledge in the field.

2.     In addition to the findings mentioned, it is essential for the authors to address the cost aspect of this process.

3.     While D-Limonene demonstrated high efficiency in dissolving ethylene vinyl acetate (EVA) and detaching the components of solar panels under various conditions with the assistance of sonication power, the economic feasibility of utilizing D-Limonene as a bio-solvent needs to be discussed.

4.     Factors such as the availability, accessibility, and cost of D-Limonene, as well as its potential scalability for large-scale industrial applications, should be considered.

5.     An analysis of the overall cost-effectiveness and potential economic benefits, in comparison to other solvents or detachment methods, would provide valuable insights for practical implementation.

Could you kindly answer these questions based on the conclusion you have drawn?

6.     How does temperature affect the separation process, particularly below 25°C?

7.     Are there any limitations or challenges associated with using low sonication power (200 W) for EVA dissolution?

8.     What are the implications of the pretreatment on EVA dissolution at different sonication powers?

9.     Are there any specific reasons why a temperature of 60°C and a contact time of 120 minutes are required for total detachment?

10.  How does the high sonication power of 700 W influence the separation process compared to lower powers?

11.  Are there any notable differences in the separation results between 60°C and 80°C at a contact time of 50 minutes?

The authors need to review and thoroughly verify the English language usage.

Reviewer 4 Report

1.The text does not provide details on the number of replicates or how the experiments were randomized. This information is crucial for evaluating the statistical significance of the results.

 2. The format and units in this paper should be unified.

 3.The text mentions the average weight of the samples as 10 g but does not provide information on the number of samples used. It is important to specify the sample size for better reproducibility.

 4.The text mentions the use of a solvent solution but does not provide details about its composition or preparation. This information is important for reproducibility and understanding the experimental procedure.

 5.The text provides an equation for the calculation of sonication power but does not explain how the values for voltage and current were obtained. It is important to provide details on how these values were measured or determined.

 6.The statement "the main reason for the dissolution of EVA is the reaction of the crosslinked and non-crosslinked part presents in the EVA film" lacks clarity and should be rephrased for better understanding.

 7.The conclusion section lacks a summary of the main findings and their significance, and it does not provide any future directions or recommendations for further research.

 This paper has some grammatical and spelling errors that need to be corrected

Round 2

Reviewer 1 Report

Thanks for the improvement 

Reviewer 3 Report

Based on the improvements you made in the manuscript, including more references, which help readers understand your topic better and relate it to existing research, I accept your work for publication. I also appreciate how you considered the cost of the process and conducted a cost-benefit analysis, showing that using D-Limonene as a bio-solvent has environmental benefits, despite being more expensive than other solvents. Your ideas for improving PV waste recycling, like better technology and larger-scale operations, are practical and useful. While we're not sure if D-Limonene can be used on a large scale yet, I commend your recognition of this uncertainty and your consideration of factors like availability and cost. Overall, I believe your manuscript will make good contribution to the field.